# Level of Acceptance of Mandatory Vaccination and Legal Sanctions for Refusing Mandatory Vaccination of Children

**DOI:** 10.3390/vaccines10050811

**Published:** 2022-05-20

**Authors:** Aneta Reczulska, Aneta Tomaszewska, Filip Raciborski

**Affiliations:** 1Faculty of Health Sciences, Medical University of Warsaw, 02-097 Warsaw, Poland; 2Department of Prevention of Environmental Hazards, Allergology and Immunology, Medical University of Warsaw, 02-091 Warsaw, Poland; anetatomaszewska@op.pl (A.T.); filip.raciborski@wum.edu.pl (F.R.)

**Keywords:** vaccination, vaccine acceptance, mandatory vaccinations, vaccination policy, children immunization

## Abstract

A preventive vaccination program is in operation in Poland. There are mandatory vaccinations for Polish residents under the age of 19 years. The law provides for financial penalties for parents who refuse to vaccinate their children. The aim of this study was to describe the attitudes of Polish residents aged 15–39 years to mandatory preventive vaccination and the level of acceptance for legal and financial sanctions for refusing mandatory vaccination of children. Materials and Methods: A face-to-face questionnaire-based study of a representative sample of 1560 residents of Poland aged 15–39 years. Data was collected in the fourth quarter of 2021. Results: In the study group, 51.5% of the respondents believed that preventive vaccination should be mandatory, and parents should have the right to decide only about additional vaccinations. Multivariate analyses (logistic regression) revealed a significant association between acceptance of mandatory vaccination and the following factors: positive COVID-19 vaccination status, self-declared religiosity, and having children. Of the 1560 respondents, 25.3% declared support for legal or financial sanctions for those refusing to vaccinate their children. In this group (*n* = 394), the highest percentage of respondents (59.4%) supported sanctions in the form of refusal to admit an unvaccinated child to a nursery or kindergarten. Conclusions: Despite preventive (mandatory) vaccination programs having been in operation in Poland since the 1960′s, only a little over 50% of adolescent Poles and young adults accept the vaccine mandate. Only 25% of this group declare their support for sanctions for refusing mandatory vaccination of children.

## 1. Introduction

Vaccination is a key prevention strategy leading to reduced incidence and mortality from infectious diseases [1,2]. An increase in the number of vaccinated individuals in a population enhances herd immunity, thus also reducing risk of disease among those who are not vaccinated [3].

The percentage of children who were not vaccinated as a result of a deliberate decision on the part of their parents increased threefold in Poland between 2015 and 2019. The child vaccination refusal rate per 1000 people was 2.3 in 2015 and 6.6 in 2019 [4,5,6,7]. The parents most often quote a fear of complications from vaccination as the reason [5].

The World Health Organization demonstrates effectiveness of preventive vaccination in eradicating the incidence of infectious diseases, such as smallpox or poliomyelitis [2]. Besides conferring individual immunity, vaccination programs determine herd (population-level) immunity by means of reducing the risk of infection among susceptible individuals owing to the presence of disease-resistant individuals [2,8,9]. According to WHO experts, expansion of the scope of preventive vaccination would result in a global decrease of the death burden by approximately 1.5 million lives annually [7].

The WHO indicates that an effective vaccine is available to control measles [10]. Measles (morbilli) is a viral infectious disease that may cause serious complications [11]. A measles vaccination is listed among mandatory vaccinations in Poland, but the incidence of measles has been growing for several years [12]. Vaccination coverage should be above 95% to ensure herd immunity, but it has been below that threshold in Poland since 2017, reaching just under 93% in 2018 [11]. A similar situation also applies to other infectious diseases. The number of cases of people with whooping cough has been rising again for several years.

Article 68 para. 4 of the Constitution of the Republic of Poland places on the authorities an obligation to combat epidemic diseases [13]. The obligation to undergo preventive vaccination in accordance with a vaccination program developed yearly by the Chief Sanitary Inspectorate is stipulated directly in Article 5 of the Act on Prevention and Control of Infection and Infectious Diseases in Humans of 5 of December 2008. This provision indicates the mandatory nature of preventive vaccination for all persons under 19 years of age in the territory of the Republic of Poland [1,14,15,16,17]. Under the Polish law, failure to comply with the vaccine mandate incurs administrative law liability [4,15,18,19]. The administrative sanction is imposed pursuant to Article 119 para. 1 of the Act of 17 June 1966 on enforcement proceedings in administration. According to the Regional Administrative Court in Warsaw, a fine has the form of forcing the entity or individual to carry out the obligation [20]. The fine does not have a punitive function insomuch as it can be imposed several times to force the entity or individual to comply with the obligation, because otherwise it would contradict the principle of ne bis in idem, i.e., nobody can be punished twice over for the same offence. One such fine may amount to PLN 10,000, or approx. EUR 2170.85 [21], but the total amount may not exceed PLN 50,000, or approx. EUR 10,800 [21,22]. Refusal to comply with the preventive vaccination mandate despite an administrative penalty represents a minor offence that may incur a fine up to PLN 1500 or approx. EUR 325.62 [21] or a reprimand. According to the law, the legal liability for refusing to have one’s child vaccinated is borne by the child’s parents or legal guardians [16,23].

In 2019, the number of warnings filed by Sanitary Inspectors was 6183, and the number of enforceable documents issued by the Chief Sanitary Inspectorate was 3397. The number of motions to regional government representatives for initiating administrative enforcement procedures was 3301 [24].

The aim of this study was to describe the attitudes of Polish residents aged 15–39 years to mandatory preventive vaccination and the level of acceptance for legal and financial sanctions for refusing mandatory vaccination of children.

## 2. Materials and Methods

### 2.1. General Description of the Study

We carried out a cross-sectional study utilizing the CAPI (Computer Assisted Personal Interview) technique and involving an original questionnaire. The respondents were interviewed at home. The study was carried out between 7 October and 10 November 2021. At the time of the study, vaccination against COVID-19 in Poland was available free of charge and voluntarily to all citizens aged 12 years and older.

### 2.2. Sampling Design

The respondents were residents of Poland aged 15–39 years. A quota random sample was selected from this population using the TERYT (the National Official Register of Territorial Division of the Country is maintained by the Central Statistical Office and regularly updated in accordance with the provisions on public statistics) sampling frame, which contains the current addresses of all households in Poland. The household addresses of study participants were drawn, taking into account the structure of administrative regions and size of towns/villages. If two or more people meeting the inclusion criteria were found in one household, the invitation to participate was issued for the latest-born individual. The resultant sample was nationally representative.

The inclusion criteria comprised residency on the territory of Poland and being 15 to 39 years old for both genders.

The exclusion criteria comprised lack of communicative command of Polish, major disability, or anticipated non-compliance with the protocol.

### 2.3. Research Tool

The research questionnaire was made up of 59 questions, of which 6 were concerned with the issue of acceptance for legal and financial sanctions for refusing mandatory vaccination of children. Other questions were concerned with the respondents’ attitudes to refusal of mandatory vaccination and the type of information that could change their decision about vaccinating their child.

The questions used in our analysis can be found in Appendix A.

### 2.4. Statistical Design

Statistical analyses used descriptive statistics and contingency tables. The level of significance of differences was determined by means of the chi-squared test. The Shapiro–Wilk test was then used to verify that data were distributed normally, and the Mann–Whitney test was used to compare medians. The analysis of the effect of individual variables on acceptance of mandatory vaccination and acceptance of legal or financial sanctions for those who refuse mandatory vaccination of their children used a multivariate logistic regression model. All analyses were performed with IBM SPSS ver. 28 (IBM Corp, Armonk, NY, USA).

### 2.5. Ethical Review

The study was acknowledged by the Bioethical Review Board at the Medical University of Warsaw (AKBE/134/2021).

## 3. Results

### 3.1. Characterisation of Study Group

Women accounted for 52.8% of the study group. The mean age was 29.01 years (SE 0.186), with the median at 30.5 (mean age of the women was 29.46, median at 31, and for the men the respective figures were 28.5 and 29.0). There was a statistically significant difference in age between genders (*p* < 0.05). Urban dwellers accounted for 57.4% of the group, with those living in agglomerations of over 500,000 residents constituting 10.8%. With regard to education, the most numerous group was formed by those with secondary education (53.2%). Having a full-time job was declared by 69.2% of the participants, and part-time or occasional jobs by another 7.4%. Those who were married constituted 51.0% of the study group, while 38.5% had children. Those declaring themselves as religious accounted for 70.9% of the group, while another 4.4% declared themselves to be deeply religious. 52.0% of the respondents had been vaccinated against COVID-19, and a further 14.0% declared that they intended to get vaccinated in the future. The detailed characteristics of the study group are shown in Table 1.

### 3.2. Attitude to Mandatory Vaccination

In the study group of 15–39-year-old residents of Poland, 51.5% of the participants believed that preventive vaccination against the most dangerous infectious diseases should be mandatory, and that parents should only have the right to decide on additional vaccination. The opposite belief (no vaccination should be mandatory, and parents should be solely responsible for deciding to vaccinate their children) was expressed by 35.1% of the respondents. The remaining participants expressed “I have a different opinion” (10.8%), or refused to answer that question (2.6%).

Gender was a statistically significant factor influencing the respondents’ opinions about mandatory vaccination against the most dangerous infectious diseases (*p* < 0.001), with 56.6% of the female participants supporting that belief, compared to 45.9% of the male participants. Age was another factor that had a statistically significant effect on the responses (*p* < 0.001), with the lowest support (43.1%) for mandatory vaccination noted in the youngest age group (15–19 years), and the highest level of support (60.5%) among the oldest participants (35–39 years). The respondents’ level of education also influenced their answers in a statistically significant manner (*p* < 0.01), with those having more education supporting mandatory vaccination more often. Being employed also influenced the respondents’ attitude in this regard (*p* < 0.05), with 54.1% of those working full-time supporting mandatory vaccination. Participants who were married declared support for mandatory vaccination considerably more often than those describing themselves as unmarried, at 59.3% vs. 44.5%, respectively (*p* < 0.001). The attitude to mandatory childhood vaccination was also influenced by the respondent having children (*p* < 0.001), as 61.2% of those who had children declared their support in this regard compared to 45.5% of respondents without children. Self-declared religiosity was also a significant factor (*p* < 0.001). Among those who described themselves as being online all the time and responding to new information and news in an ongoing fashion, 47.3% declared their support for mandatory vaccination against the most dangerous diseases, compared to 59.7% of those who declared that they used the Internet only to perform specific tasks. Support for mandatory vaccination was expressed by 27.2% of those declaring themselves as non-religious, and 66.2% of those profoundly religious. Support was also significantly dependent on a positive COVID-19 vaccination status (*p* < 0.001), with 65% of those declaring that they had been vaccinated against COVID-19 supporting mandatory vaccination, compared to 30.2% of those unvaccinated against COVID-19 and not intending to get vaccinated.

The size of the population at the place of residence was shown not to significantly influence support for mandatory vaccination against the most dangerous infectious diseases (*p* = 0.103). The detailed results are shown in Figure 1.

Out of all participants, 25.3% support legal or financial sanctions for those who refuse mandatory vaccination of their children (including 2.8% declaring complete support, and 22.5% saying they would moderately support that move). The responses were affected by gender (*p* < 0.05), with higher levels of support among women (27.6% vs. 22.7%), as well as by age (*p* < 0.01), with the levels of support for penalizing refusal to vaccinate children growing with advancing age of the respondents from 17.7% among those aged 15–19 years to 30.1% among those 35–39 years old. The population size of respondents’ places of residence was also a factor (*p* < 0.05), with the highest levels of support for penalizing avoidance of vaccination for children noted among those living in cities with populations between 50,000 and 100,000 (34.1%) and populations above 500,000 (32.7%), and the lowest levels noted in towns with populations between 20,000 and 50,000 (19.6%) and in rural areas (23.3%). Those who were better educated were more likely to accept penalties for refusing vaccination of children than less well-educated respondents (*p* < 0.05), with 27% of those with a university education supporting such moves vs. 15.8% of those with primary or junior secondary education. Being employed also influenced respondents’ attitudes towards penalties for avoidance of vaccination of children (*p* < 0.001), with 28.1% of those employed full-time supporting the penalization, vs. 16.7% of those respondents working part-time (or occasionally) and 19.8% of those who were not employed. Marital status and having children also influenced respondents’ attitude towards legal or financial sanctions against those who avoid mandatory vaccination of their children (*p* < 0.001 for both variables), where respondents having children were more likely to accept penalization of avoidance of child vaccination than those without children (32.0% vs. 21.0%). Self-declared religiosity also significantly influenced respondents’ views (*p* < 0.001), with 55.9% of those declaring to be profoundly religious accepting penalization of avoidance of child vaccination compared to 10.7% of those defining themselves as completely non-religious. Having been vaccinated against COVID-19 also had a significant effect on respondents’ opinion about the penalization (*p* < 0.001), with 37.9% of those vaccinated accepting the penalization vs. 6.7% of those not vaccinated and not intending to get vaccinated.

Internet use did not influence respondents’ opinions regarding the penalization of avoidance of mandatory child vaccination (*p* = 0.321). The detailed data can be found in Figure 1.

Among those who wholly or moderately support the use of legal or financial sanctions for vaccine refusal, (*n* = 394), the largest percentage (59.4%) support measures in the form of refusal to admit an unvaccinated child to a nursery or kindergarten, while 43.9% would agree to forbidding unvaccinated children from taking part in organized holiday trips (summer/winter camps). A similar percentage (43.7%) would accept prohibiting access of unvaccinated children to additional classes and activities at state-owned institutions. Fines were accepted by 27.7% of the respondents supporting penalization of those refusing to vaccinate their children, while forced vaccination was regarded as admissible by 21.1% of respondents from this group. The detailed data are shown in Figure 2. Having children did not significantly influence the replies, except for the attitude to forced vaccination (*p* < 0.05), which was supported by 16.1% of the respondents having children and 25.7% of those without children.

A multivariate logistic regression model of the impact of individual variables on acceptance of mandatory vaccination of children against the most dangerous diseases revealed a Cox and Snell R^2^ goodness-of-fit index of 0.135, and a Nagelkerke R2 index of 0.181. The strongest effects were noted for self-declared religiosity and positive COVID-19 vaccination status. Those who declared themselves to be profoundly religious had more than three times higher odds of supporting mandatory vaccination (OR = 3.31; 95%CI 1.64–6.71) than those identifying themselves as completely non-religious. Respondents declaring themselves to be religious demonstrated similar levels of support (OR = 2.28; 95%CI 1.41–3.68). The odds ratio for supporting mandatory vaccination was also more than 3.5 times higher among those declaring themselves to have been vaccinated against COVID-19 (OR = 3.69; 95%CI 2.88–4.74) over those not vaccinated and not intending to get vaccinated. Those who were unvaccinated but intending to get the COVID-19 vaccine had 2.5 times higher odds (OR = 2.52; 95%CI 1.81–3.53) of supporting mandatory vaccination. Having children did not significantly influence these results, but there was one exception, namely, respondents with children aged 7–15 years had 42% higher odds (OR= 1.42; 95%CI 1.07–1.88) of supporting mandatory child vaccination compared to those not having children in this age group. Furthermore, those respondents who went online only in order to perform specific tasks displayed 39% higher odds (OR= 1.39; 95%CI 1.1–1.77) of accepting mandatory vaccination compared to those who described themselves as being online all the time. The other variables of interest did not exert a statistically significant effect. These results are shown in Table 2.

A multivariate logistic regression model of the impact of individual variables on the acceptance of legal or financial sanctions towards those who refuse mandatory vaccination of their children showed a goodness of fit index of 0.148 according to Cox and Snell R^2^ and of 0.219 according to Nagelkerke R2. The strongest association between acceptance of penalization of refusal to vaccinate children and the factors analyzed was noted for declared COVID-19 vaccination status. Those who declared to have been vaccinated had nearly eight times higher odds (OR = 7.84; 95%CI 5.31–11.56) of accepting legal or financial sanctions compared to those declaring not having been vaccinated and not intending to do so in the future. Those respondents who had not been vaccinated but intended to get vaccinated against COVID-19 displayed more than 4.5 times higher odds (OR = 4.6; 95%CI 2.86–7.39) of supporting the penalization compared to those not intending to get vaccinated. Those describing themselves as profoundly religious demonstrated nearly 6.5 times higher odds (OR = 6.39; 95%CI 2.72–14.97) of accepting the penalization compared to those declaring themselves to be completely non-religious. Furthermore, the parents of children up to 6 years of age were also more likely to accept legal or financial penalties for individuals refusing to vaccinate their children, for 42% higher odds (OR = 1.42; 95%CI 1.04–1.94) than those who did not have children in that age group. The other variables in the model did not influence the results in a statistically significant manner. The detailed results are presented in Table 2.

## 4. Discussion

### 4.1. Key Results

Young (15–39 years old) residents of Poland have mixed opinions regarding mandatory vaccination. The dominant (51.5%) view is that preventive vaccination against the most dangerous infectious diseases should be mandatory; at the same time, however, more than a third of the respondents (35.1%) support complete freedom to (not) vaccinate children. A more extensive analysis utilizing a multivariate logistic regression model demonstrated that respondents’ attitudes towards mandatory vaccination were strongly influenced by self-declared religiosity and COVID-19 vaccination status, with those who more religious and those vaccinated against COVID-19 more likely to support mandatory vaccination. Moderate use of the Internet and having a child aged 7–15 years were also associated with acceptance of mandatory vaccination.

Only a fourth of the respondents (25.3%) declared support for legal or financial sanctions towards those refusing mandatory vaccination of their children. Additionally, a considerable majority of the respondents declared moderate support. The logistic regression model showed that, also in this case, those deeply religious and those vaccinated or intending to get vaccinated against COVID-19 were more likely to support penalizing the refusal to vaccinate children. A similar pattern was revealed for parents of children up to 6 years of age. Of the available options of sanctions against those refusing to vaccinate their children, it was the refusal to admit an unvaccinated child to a state-owned nursery or kindergarten that was the most popular measure.

### 4.2. Limitations

The most important limitation in this study is that the results are based on declarations on the part of the respondents, which may differ from their actual views or actions. The declarations regarding COVID-19 vaccination status of the respondents or their immediate families were not verified in any way.

### 4.3. Interpretations

Misgivings regarding vaccine safety and effectiveness have also been reported by other researchers in Poland. A 2017 study by A. Włodarska et al., found that 73% of those polled were convinced that vaccines are safe, while 12.5% did not believe that vaccines are effective [25]. In 2019, Szalonka demonstrated that 68% respondents were very much or rather afraid of the medical sequelae of vaccination [5]. This trend has also been observed in other countries, e.g., in Italy. Brunelli et al., found in their 2020 study that 38.8% of respondents were afraid of potential adverse drug reactions [26]. The same motivation has been revealed in other studies [7,27]. Offit et al., in the US [28] and Heiniger et al., in Germany [29] found that parents believed that children’s immune systems were overloaded due to vaccinations.

Our study focused on the opinions of the public regarding mandatory preventive vaccination and acceptance of financial penalties and legal consequences of refusal to vaccinate children. However, the respondents were split into two groups. One represented the proponents of preventive vaccination who accepted the mandatory vaccination scheme (51.5%), while the other group stated that vaccination should be optional and decisions to vaccinate taken by a child’s parents (35.1%). The remaining respondents had a different opinion. Similar results were obtained by Szalonka in a 2020 questionnaire-based study, where 45% of respondents agreed with the rationale behind establishing a system of mandatory preventive vaccination, with some (40%) believing that this arrangement was not necessary [30].

Our study revealed that the level of acceptance of and support for mandatory vaccination of children was significantly influenced by declared religiosity. Other studies have yielded similar conclusions [31,32]. We have revealed that the time spent surfing the Internet affects attitudes towards vaccines. In the Szalonka’s 2020 study [32], the influence of Internet use on the attitude of the respondents was also found. In 2019, Duda et al., found that parents of unvaccinated children indicated online resources as the main source of knowledge about preventive vaccination [33]. Parents’ refusals to vaccinate children may be associated with fake information disseminated on online social media [34].

A 2017 study by Mathieu found parents’ level of education and income as key factors for acceptance of preventive vaccination [35]. In 2018, Warakomska and Walińska demonstrated that education as well as professional activity exerted a significant effect on parents’ attitude to mandatory vaccination. These data may testify to a lack of knowledge among parents about preventive vaccination and possible adverse drug reactions [7]. Our study used a multivariate logistic regression model and found that parents’ level of education did not influence their attitudes towards vaccination and towards penalties and legal consequences.

Penalties and legal sanctions for those refusing mandatory vaccination of their children was accepted by as few as 25.3% of the respondents. The 2018 Australian study by Helps found that financial penalties were not a sufficient policy to effect a change of decision among those refusing to vaccinate their children [36]. Conversely, an Italian study by Casul in 2021 reported higher vaccination rates among children in the wake of the 2017 decree of Lorenzin stipulating mandatory vaccination, with unvaccinated children nunable to attend nurseries and kindergarten and the parents liable to pay fines [37].

More than half of the respondents (59.4%) supporting financial penalties and legal sanctions for vaccine refusers declared support for consequences in the form of unvaccinated children not being allowed to attend a state-owned nursery or kindergarten. The European tribunal of Human Rights has regarded the refusal to allow an unvaccinated child to attend kindergarten a wholly legal means with a preventive rather than punitive function [38].

Other studies have indicated that there are suggestions that those refusing vaccination for their children should cover the cost of treatment if the child gets infected [27,39]. This would be a less aggressive form of penalization, a sanction against the negative societal consequences of refusal of vaccination.

## 5. Conclusions


Even though a mandatory vaccination scheme has been operating in Poland since the 1960s, the opinions about mandatory vaccination are divided. Only more than half of those aged 15–39 years are in favor of mandatory vaccination.One in four respondents declared support for legal or financial sanctions against those refusing mandatory vaccination of their children.The most widely accepted sanctions for refusing to vaccinate children include the refusal to admit an unvaccinated child to a nursery or kindergarten, refusal to allow such children to go on organized holiday trips, and refusal to allow such children to take part in extracurricular activities offered by state-owned institutions.More extensive statistical analyses based on multivariate models revealed no effect of most typical sociodemographic variables (e.g., gender, education, size of population in place of residence) on respondents’ attitudes to mandatory vaccination and penalization of vaccination refusal. Significant factors comprised self-declared religiosity and a positive COVID-19 vaccination status.


## Figures and Tables

**Figure 1 vaccines-10-00811-f001:**
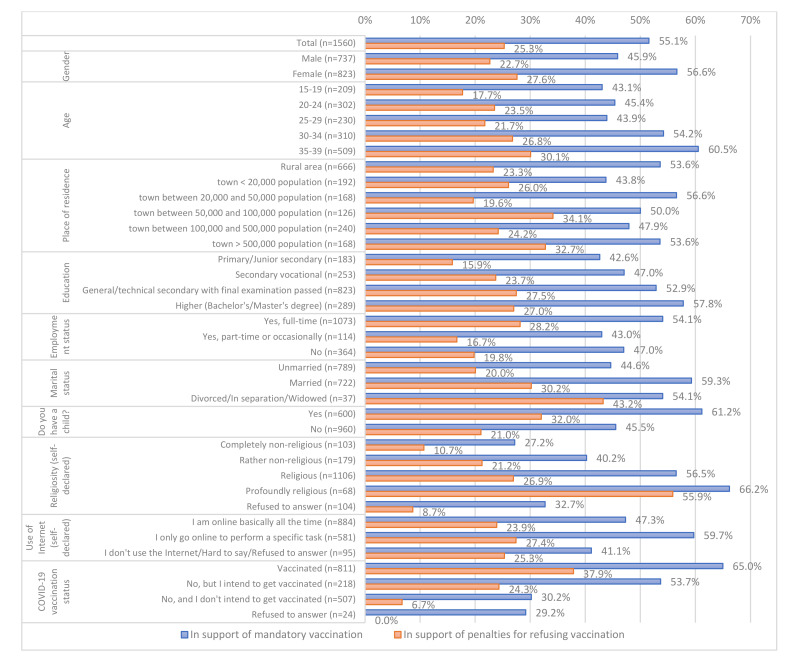
The percentages of respondents declaring that preventive vaccination against the most dangerous infectious diseases should be mandatory and the percentages of respondents accepting legal or financial sanctions against those who refuse mandatory vaccination of their children (*n* = 1560).

**Figure 2 vaccines-10-00811-f002:**
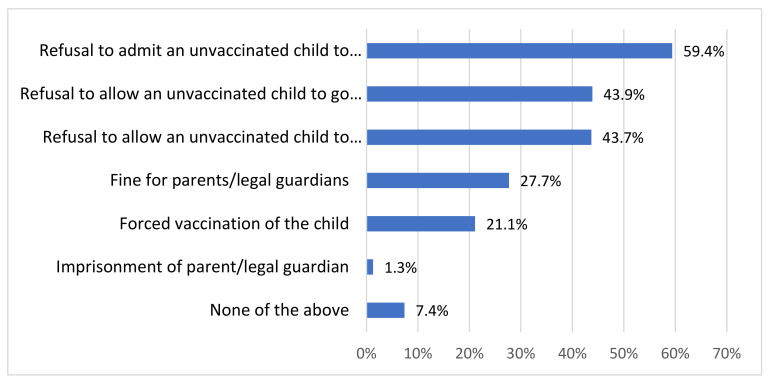
The percentages of respondents admitting specific sanctions against individuals refusing mandatory vaccination of their children. This question was asked only of those respondents who had stated that they would accept sanctions against vaccine refusers (*n* = 394).

**Table 1 vaccines-10-00811-t001:** Characteristics of the study group.

	*n*	%
Total	1560	100
**Gender**		
Male	737	47.2
Female	823	52.8
**Age**		
15–19	209	13.4
20–24	302	19.4
25–29	230	14.7
30–34	310	19.9
35–39	509	32.6
**Place of residence**		
rural area	666	42.7
town < 20,000 population	192	12.3
town between 20,000 and 50,000 population	168	10.8
town between 50,000 and 100,000 population	126	8.1
town between 100,000 and 500,000 population	240	15.4
town > 500,000 population	168	10.8
**Education**		
Primary/Junior secondary	183	11.8
Secondary vocational	253	16.3
General/technical secondary with final examination passed	823	53.2
Higher (Bachelor’s/Master’s degree)	289	18.7
**Employment status**		
Yes, full-time	1073	69.2
Yes, part-time or occasionally	114	7.4
No	364	23.5
**Marital status**		
Unmarried	789	51
Married	722	46.6
Divorced/In separation/Widowed	37	2.4
**Do you have a child?**		
Yes	600	38.5
No	960	61.5
**Religiosity (self-declared)**		
Completely non-religious	103	6.6
Rather non-religious	179	11.5
Religious	1106	70.9
Profoundly religious	68	4.4
Refused to answer	104	6.7
**COVID-19 vaccination status**		
Vaccinated	811	52
No, but I intend to get vaccinated	218	14
No, and I don’t intend to get vaccinated	507	32.5
Refused to answer	24	1.5

**Table 2 vaccines-10-00811-t002:** Influence of selected factors on support for mandatory vaccination against the most dangerous infectious diseases. A multivariate logistic regression model (*n* = 1560).

		Mandatory Vaccination		Penalties for Refusing Vaccination	
	*n*	Sig.	OR (95% CI OR)	Sig.	OR (95% CI OR)
**Gender**					
Male	737	Ref.	Ref.	Ref.	Ref.
Female	823	0.145	1.18 (0.94–1.48)	0.896	1.02 (0.78–1.32)
**Age**					
15–24 years	511	Ref.	Ref.	Ref.	Ref.
25–39 years	1049	0.981	1 (0.72–1.39)	***p* < 0.05**	**0.67 (0.45–0.98)**
**Place of residence**					
rural area	666	Ref.	Ref.	Ref.	Ref.
town < 50,000 population	360	0.447	0.90 (0.68–1.19)	0.865	1.03 (0.74–1.43)
town between 50,000 and 100,000 population	126	0.588	0.89 (0.59–1.36)	0.055	1.57 (0.99–2.49)
town between 100,000 and 500,000 population	240	0.221	0.82 (0.59–1.13)	0.708	1.08 (0.74–1.57)
town > 500,000 population	168	0.553	0.89 (0.62–1.3)	0.162	1.35 (0.89–2.05)
**Education**					
Primary/Junior secondary	183	Ref.	Ref.	Ref.	Ref.
Secondary vocational	253	0.887	0.97 (0.6–1.56)	0.358	1.33 (0.73–2.42)
General/technical secondary with final examination passed	823	0.572	1.13 (0.75–1.7)	0.332	1.3 (0.77–2.21)
Higher (Bachelor’s/Master’s degree)	289	0.519	1.18 (0.72–1.93)	0.718	1.12 (0.61–2.06)
**Employment status**					
Yes, full-time	1073	0.429	0.87 (0.61–1.24)	0.338	1.23 (0.81–1.86)
Yes, part-time or occasionally	114	0.27	0.76 (0.47–1.24)	0.288	0.71 (0.38–1.33)
No	364	Ref.	Ref.	Ref.	Ref.
**Having children**					
children 0–6 years old = NO	1241	Ref.	Ref.	Ref.	Ref.
children 0–6 years old = YES	319	0.684	0.94 (0.71–1.25)	***p* < 0.05**	**1.42 (1.04–1.94)**
children 7–15 years old = NO	1185	Ref.	Ref.	Ref.	Ref.
children 7–15 years old = YES	375	***p* < 0.05**	**1.42 (1.07–1.88)**	0.125	1.27 (0.94–1.73)
children 16 years old and older = NO	1513	Ref.	Ref.	Ref.	Ref.
children 16 years old and older = YES	47	0.411	1.32 (0.68–2.56)	0.701	0.87 (0.42–1.78)
**Religiosity (self-declared)**					
Completely non-religious	103	Ref.	Ref.	Ref.	Ref.
Rather non-religious	179	0.156	1.5 (0.86–2.63)	0.145	1.77 (0.82–3.8)
Religious	1106	***p* < 0.001**	**2.28 (1.41–3.68)**	0.054	1.95 (0.99–3.86)
Profoundly religious	68	***p* < 0.001**	**3.31 (1.64–6.71)**	***p* < 0.001**	**6.39 (2.72–14.97)**
Refused to answer	104	0.631	1.17 (0.62–2.19)	0.451	0.69 (0.26–1.82)
**Use of Internet (self-declared)**					
I am online basically all the time	884	Ref.	Ref.	Ref.	Ref.
I only go online to perform a specific task	581	***p* < 0.01**	**1.39 (1.1–1.77)**	0.603	1.08 (0.82–1.42)
I don’t use the Internet/Hard to say/Refused to answer	95	0.301	0.78 (0.49–1.25)	0.559	1.18 (0.68–2.02)
**COVID-19 vaccination status**					
Vaccinated	811	***p* < 0.001**	**3.69 (2.88–4.74)**	***p* < 0.001**	**7.84 (5.31–11.56)**
No, but I intend to get vaccinated	218	***p* < 0.001**	**2.52 (1.81–3.53)**	***p* < 0.001**	**4.6 (2.86–7.39)**
No, and I don’t intend to get vaccinated	531	Ref.	Ref.	Ref.	Ref.

## Data Availability

The datasets generated during and/or analyzed during the current study are available from the corresponding author on reasonable request.

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
