# Peer review of "Level of Acceptance of Mandatory Vaccination and Legal Sanctions for Refusing Mandatory Vaccination of Children"

_vaccines, 2022, doi:10.3390/vaccines10050811_

Round 1

Reviewer 1 Report

[Vaccines] Manuscript ID: vaccines-1682711

This paper presents attitudes of Polish residents aged 15 to 39 about mandatory vaccination and the level of acceptance of financial (or other) sanctions against parents who refuse to vaccinate their children. In a sample of 1,560 persons, the authors found that 51.5% were in favour of mandatory preventive vaccination, while 25.3% declared that they supported legal and/or financial sanctions for non-compliance. The most widely acceptable form of punishment would be refusal to accept unvaccinated children in state-run childcare structures (nursery/kindergarten).

Overall, the findings are interest and the paper is well written. However, I have a few questions. Firstly, what is the rationale for choosing the age group of 15 to 39 years? Firstly, the younger end of this spectrum (adolescents, aged 15 – 18, the official age of adulthood) are too young to have their own children, so are likely unconcerned by the preventive vaccination mandate. They may still be eligible to receive some of the vaccines covered by the mandate. So can adolescents of this age go for vaccination on their own, without parental consent? I’m not sure why this age group is included in this survey. At the opposite end of the spectrum, I feel that the cut-off of 39 years of age may be too young. Many adults in their forties and fifties still have young children and adolescents in their household, and are still concerned by the question of vaccination. So why were they excluded? If the age group sampled had been 25 to 50, the results may have been significantly different. Similarly, only 38.5% of the study population had children, yet having children in certain age groups was significantly associated with the outcome. The same survey in a sample of ONLY people with children might yield different results. I think the authors need to justify these choices in more detail. Just because the sample size is representative of the overall population structure does not mean that it is representative of the group targeted by the measures being investigated.

My second question is more existential: Given that the mandate and the provisions for sanctions already exist in Polish law, what does it matter whether people agree or not? It is already the law, so regardless of whether the population accepts the measure or complies with it, the law exists nevertheless. This type of study is interesting to guide policy when the introduction of new measures is being considered. But insofar as these measures are already enshrined in the law (and since the 1960s), what are the perspectives from this research? What do the authors hope to gain or change by showing that only half of those sampled agree?

Some suggestions for minor corrections:

  • Page 2, lines 49 to 84: There is slightly too much detail about the legal provisions and sanctions. I think it would suffice to cite the dispositions of the law, and the potential sanctions. Conversely, the authors should specify the list of vaccines covered by the mandate, and give some information about current levels of uptake of these guidelines. The paragraph on measles is too long and detailed, and there is no reason to detail measles in this way but not the other vaccines. However, the idea is OK – keep some (but less) information about measles, but complement it with similar data for the other mandatory vaccines.
  • Page 3, line 109, there is a mention of Appendices 3 and 4, but there was no mention before this of Appendices 1 and 2.
  • Page 3, line 126, in the Results: The authors state that “Gender had a statistically significant effect on age” – this is poorly stated, your gender cannot change your age! I think what you mean to say that that there was a statistically significant difference in age between genders. Please reformulate (and perhaps check for the same phrasing elsewhere in the article).
  • In terms of visual attractiveness, Figure 1 is almost illegible. I think this data would be more suited to a Table format. Perhaps the authors could try presenting it as a table, to see if it is more palatable.
  • Page 6, line 213 – I fail to see the utility of sanctioning parents by refusing unvaccinated children on package holidays. Package holidays are a leisure activity, at the personal discretion of the parents, and it is unlikely that commercial tour operators could be convinced to put public health considerations before financial profit by refusing to accept unvaccinated children on their package holidays. Penalizing parents for a failure to comply with public health measures by interfering with their private life (via third-party industry) is neither feasible nor ethically tenable. Public health compliance should be required for domains that remain under the authority of the state, such as schools, childcare structures (financed in any way by the state), municipal buildings and activities, and possibly also by bearing the cost of recourse to public services (e.g. making parents pay if the child gets sick with the target disease).
  • Page 9, lines 318-319 – the authors state that reliance on the internet as a source of information about preventive vaccinations was a key influence. However, the results provided do not support this. I did not understand from the results that the questions about internet used questioned the participants about what they were doing on the internet. I don’t think the authors can affirm whether it was the type of information taken from internet that forged their opinions; this is over-interpretation. They should limit the interpretation to the strict finding, namely that those who spend more time on the internet was related to vaccination attitudes. You don’t know for sure what people were doing online.
  • Page 1, line 41: I think it should read “eradicating OR minimising” (not “of”)
  • Page 1, line 42, the authors should put “smallpox” instead of “variola vera”
  • Page 5, line 172, spelling error at the last word (significantly, not sifnificantly)

Reviewer 2 Report

Authors report significant effect of self-declared religiosity and a positive COVID-19 vaccination status on respondents’ attitudes toward mandatory vaccination and following measures in Poland. Despite the limitations of the current study such as lack of verification for COVID-19 vaccination and declaration-based survey, the results presented provide meaningful insight into people's thought on mandatory vaccination in Poland. 

Figure 1 is difficult to read. Either the font should be increased or the format be changed for better reading.

The grammatical errors or typos must be corrected before publication. Some of them are listed below.

line 42: eradicating of minimising -> eradicating or minimizing

line 61: ... of 5 of December 2008 -> must be checked for correct usage.

line 227 R2index of 0.181. -> R2 index of 0.181.

Author Response

Thank you very much for your comments and for taking the time to review our work. The comments have been implemented in the text. The following items have been corrected:

-page 1, line 41, "eradicating of minimising" changed to “eradicating”

-line 227- changed to R2 indicator

-grammatical errors and typos were corrected in the text.

Reviewer 3 Report

This manuscript explores an important topic related to methods to encourage childhood vaccination. Investigating this topic is particularly timely as perspectives about vaccination are being impacted by the COVID-19 pandemic. This manuscript therefore provides valuable information. However, in some areas, not enough information is shared regarding the methods of this work. Furthermore, little discussion is dedicated to the timing of this work in relation to the COVID-19 pandemic. Please see below for questions and suggestions that may be addressed to improve this manuscript.

Abstract:

Please include the time frame of data collection in the Abstract.

Introduction:

The authors state, “The percentage of children who were not vaccinated as a result of a deliberate decision on the part of their parents increased threefold in Poland between 2015 and 2019.” Can the authors provide the actual percentages related to this statement to give an idea of how prevalent this refusal to vaccinate is in Poland?

The legal/policy information is provided in a level of detail that exceeds the scope of this work. The authors may consider condensing this section of the Introduction to explain the aspects of the legal policies that are most pertinent to this study.

Some information related to the legal policies, however, are missing. How are these legal fines administered or enforced? Are they enacted yearly? Per child or per family?

Furthermore, many terms are used to describe the sanctions/penalties surrounding refusal to vaccinate. What is the difference between a fine, administrative fine, minor offense fine, or reprimand? Please be consistent with terms and define them accordingly as you use them.

The authors state, “In 2019, the number of warnings filed by Sanitary Inspectors was 6,183, and the number of enforceable documents issued by the Chief Sanitary Inspectorate was 3,397.” Can the authors include an estimate of the number of children in Poland to give a sense of the prevalence of these actions? What is a warning and how does it differ from an enforceable document or a motion for initiating administrative enforcement? Please simplify the legal language, if possible.

Materials and Methods:

Please explain why the age range of 15-39 years was chosen. It would seem that including individuals 15-19 years of age would complicate the results of this study, in that individuals in this age range will have perspectives both as individuals within the age range for which vaccination is mandatory but may also have perspectives related to that of an adult/parent. How are these perspectives parsed in this work and why did the authors decide to include individuals under age 19 for this study?

Please explain the availability of COVID-19 vaccinations in Poland during the timeframe of data collection.

Please define, describe, and cite the TERYT sampling frame.

Please provide additional information about the study design. Were incentives provided for study participation? How were eligible participants approached? How many attempts were made to each randomly selected individual (e.g., if they were not home)? Who administered the survey face-to-face in individual’s homes? How did you reduce the chance of potential coercion or bias? Could individuals refuse to answer questions or were they provided an option to record no response? How were survey answers recorded (e.g., paper, electronically)? How was missing data handled in the analyses?

At the very least it is critical to describe in the text the scales used for the outcome variables of support for mandatory vaccines and penalties. Were these based on a likert scale? Yes/no options? How was support vs. non-support derived from these answers?

Results:

The number of significant digits to report age means and SE is inconsistent and likely excessive. Reporting to one decimal place is likely sufficient.

The authors state “Gender had a statistically significant effect on age (p<0.05).” I don’t believe this comparison is necessary.

The survey asked participants about their beliefs regarding mandatory preventative vaccination against the most dangerous infectious diseases. What diseases does this include? Were these diseases defined for the participants or was this left open for interpretation? Did this include COVID-19 vaccination?

Sometimes the authors state that participants “expressed a different opinion.” In these instances, can the authors please clarify what other response options the participants picked or if they declined from answering the question?

Please consider replacing “better educated” with “having more education” or some other alternative.

The analysis related to internet use is unexpected as this variable is not discussed at all in the introduction or methods. What is the justification for looking at this as a variable? More information is needed to integrate this into the study design. Furthermore, the presentation of results on internet use is awkward as it interrupts the reporting of results related to religiosity.

In Figure 1, the legend should be “in support of mandatory vaccination” and “in support of penalties for refusing vaccination” or some equivalent. The open bars are difficult to see. Can filled bars be used instead? Using “.” or “,” for decimals should be made consistent between the text and figures. Rural area should not be in the age category.

What is a package holiday?

If the authors choose to report goodness of fit for logistic regression, please provide interpretation of these results, and consider how they are meaningful to the interpretation of this data.

For Table 2, the title should address both support for mandatory vaccination and for penalties for refusing vaccination. Please define what bolding means and report actual P-values, unless <0.001. Please note that the given categories of children’s ages currently exclude those children 6 years of age (they fit neither in the <6 or 7-15 groups).

The authors do not mention the analysis by children’s age until they are reported in Table 2. Please include discussion of this analysis earlier in the paper and consistently report analyses either by having children only (e.g., yes/no) or by having children in certain age groups.

Discussion:

The results of the study are reported excessively again in the discussion and then again in the conclusion. The authors should focus on summarizing their results briefly and discussing them in the context of the existing literature and landscape of the study timing.

The authors state, “Our study focused on the opinions of parents or legal guardians of children regarding mandatory preventive vaccination and acceptance of financial penalties and legal consequences of refusal to vaccinate children.” This statement is not true, in that the study population included those that were not parents or legal guardians.

The authors state that their findings align with previous research. This provides valuable insight into the agreement with other evidence, but also calls into question the added value of this study. What about this study adds to the current literature? One can imagine that the timing of this work is particularly prudent, due to the impact of COVID-19 on many perspectives. Can the authors discuss this aspect of the study in more detail?

How has the COVID-19 pandemic impacted childhood vaccination in Poland? It would be helpful to discuss opinions about childhood COVID-19 vaccination and explain if this is included in the “most dangerous diseases” covered by the mandatory childhood vaccines.

Can the authors please clarify if there are currently penalties in Poland such as those that prevent unvaccinated children from attending daycare or kindergarten?

Can the authors please discuss if there is any evidence or discussion of (for or against) how effective these strategies (mandates/penalties) are for increasing adherence to childhood vaccination?

Is there a global comparison to be made about hesitancy toward childhood vaccination? For example, in the US some hesitancy stems from concerns about the link between vaccinations and autism. What similarities or differences are there regarding potential motivations or hesitancy toward childhood vaccination in different nations?

The authors may consider more deeply the limitations of this study. Was there potential for bias in the way the survey was administered? Were there limitations in how the questions and responses were prepared that limited the insight into individuals’ perspectives on these topics?

Some of the references have very little information or are not presented in English. Are these references accessible to all readers as a resource? If not, perhaps different/additional information or references are needed.

Finally, please proofread the manuscript to edit for English language mistakes and typos, define acronyms, remove extraneous text, and check consistency of reporting significant digits and decimal points.

Round 2

Reviewer 1 Report

The authors have addressed the points raised.